# An Adaptive Threshold-Based Pixel Point Tracking Algorithm Using Reference Features Leveraging the Multi-State Constrained Kalman Filter Feature Point Triangulation Technique for Depth Mapping the Environment

**DOI:** 10.3390/s25092849

**Published:** 2025-04-30

**Authors:** Zohaib Wahab Memon, Yu Chen, Hai Zhang

**Affiliations:** School of Automation Science and Electrical Engineering, Beihang University, Beijing 100191, China; zy2203101cy@buaa.edu.cn

**Keywords:** SLAM, VIO, MSCKF, optical flow, reference features, tracking, depth map

## Abstract

**Highlights:**

Images consist of tens of thousands of pixels. With limited computational power, only the pixels with features can be tracked across sequential images using either optical flow or descriptors, which cannot guarantee a complete depth map of the environment. This article proposes a method by which a depth map of the pixels (even those without features) in an image can be generated by tracking pixel points using reference features in the neighborhood and estimating depth with the MSCKF-VIO feature point triangulation method.

**What are the main findings?**

**What are the implications of the main findings?**

**Abstract:**

Monocular visual–inertial odometry based on the MSCKF algorithm has demonstrated computational efficiency even with limited resources. Moreover, the MSCKF-VIO is primarily designed for localization tasks, where environmental features such as points, lines, and planes are tracked across consecutive images. These tracked features are subsequently triangulated using the historical IMU/camera poses in the state vector to perform measurement updates. Although feature points can be extracted and tracked using traditional techniques followed by the MSCKF feature point triangulation algorithm, the number of feature points in the image is often insufficient to capture the depth of the entire environment. This limitation arises from traditional feature point extraction and tracking techniques in environments with textureless planes. To address this problem, we propose an algorithm for extracting and tracking pixel points to estimate the depth of each grid in the image, which is segmented into numerous grids. When feature points cannot be extracted from a grid, any arbitrary pixel without features, preferably on the contour, can be selected as a candidate point. The combination of feature-rich and featureless pixel points is initially tracked using traditional techniques such as optical flow. When these traditional methods fail to track a given point, the proposed method utilizes the geometry of triangulated features in adjacent images as a reference for tracking. After successful tracking and triangulation, this approach results in a more detailed depth map of the environment. The proposed method has been implemented within the OpenVINS environment and tested on various open-source datasets supported by OpenVINS to validate the findings. Tracking arbitrary featureless pixel points alongside traditional features ensures a real-time depth map of the surroundings, which can be applied to various applications, including obstacle detection, collision avoidance, and path planning.

## 1. Introduction

Autonomous visual–inertial integrated navigation systems (VINS) have gained widespread adoption, especially in Global Navigation Satellite System (GNSS)-denied environments, such as indoors, underwater, in urban canyons, and in space [1]. Most inertial navigation systems (INS) rely on inertial measurement units (IMU) to measure the platform’s local linear acceleration and angular velocity. However, high-end tactical-grade IMUs are commercially unviable; thus, low-cost, lightweight micro-electromechanical (MEMS) inertial measurement units (IMUs) are utilized. The MEMS-based IMUs are affected by noise and bias, making the simple integration of IMU measurements to estimate the 6DOF pose (position and orientation) unreliable for long-term navigation. When GNSS signals are available, the motion drift accumulated over time due to IMU manufacturing imperfections can be corrected using the global information from GNSS [2]. However, in GNSS-denied environments, a monocular camera that is lightweight, small in size, and energy-efficient, and provides substantial environmental data, can be utilized to correct the drift accumulated in the navigation solution, yielding visual–inertial integrated navigation systems (VINS) [3].

Theoretically, visual–inertial navigation can be classified as either visual–inertial simultaneous localization and mapping (VI-SLAM) or visual–inertial odometry (VIO) [3]. VI-SLAM performs localization and environmental mapping simultaneously, which makes it computationally expensive. In contrast, VIO is a localization method that does not perform environmental mapping; instead, it utilizes environmental features such as points, lines, and planes to correct the navigation solution affected by IMU imperfections. Since VIO does not perform environmental mapping, it estimates only a limited number of environmental features to refine the navigation solution, making it computationally efficient compared to VI-SLAM; however, this efficiency comes at the cost of losing mapping information.

Like SLAM systems, all visual–inertial navigation systems, including VIO and VI-SLAM, can be divided into front-end and back-end components. The front end processes the visual information, and the back end serves as the estimation engine [4]. In short, we can say that the front end performs image processing or computer vision tasks, such as the extraction and tracking of environmental features. The back end deals with the estimation process, which is performed in various ways, primarily through loosely coupled and tightly coupled estimation methods. The loosely coupled methods independently process the visual and inertial measurements [5]. In contrast, the tightly coupled methods derive constraints between the visual and inertial measurements to obtain the motion and observation models [6]. The loosely coupled and tightly coupled methods are further classified as filtering-based [5,6] or optimization-based [7]. The optimization approach is more efficient but computationally expensive because it solves a non-linear least squares problem over a set of measurements through re-linearization. Research in this area is also progressing with the emergence of several optimized, computationally efficient, and open-source libraries, such as Ceres [8] and g2o [9]. On the other hand, filtering-based back ends are still popular among researchers because of their computational efficiency [3,4].

Among the tightly coupled filtering methods of VIO, the multi-state constrained Kalman filter (MSCKF) [10] developed by Mourikis and Roumeliotis is a masterpiece based on the extended Kalman filter (EKF) [11], which maintains a sliding window of IMU/camera poses and does not estimate environmental features in the state vector, as opposed to SLAM. Instead, a static environmental feature observed by multiple camera poses is triangulated and used for measurement updates. State propagation is performed using quaternion-based inertial dynamics with the IMU measurements, and state updates are performed by projecting the visual bearing measurements onto the left null space of the feature Jacobian, constraining the motion to the cloned IMU/camera poses. Thus, the computational cost associated with estimating thousands of feature points is eliminated.

The MSCKF algorithm has been extended in several directions, notably the stereo cameras in the fast UAV autonomous flight [12], LARVIO [13], have extended MSCKF to include zero-velocity updates, and OpenVINS [14] has combined the MSCKF with a VIO and SLAM framework, incorporating online camera calibration, zero velocity detection, static and dynamic state initialization, etc. All three studies have open-sourced their implementations and tested them on several open-source public datasets, such as EuRoC [15].

All these studies have focused on VIO [12,13,14], specifically localization. A complete and robust visual–inertial navigation system must offer adequate information for applications like path planning, collision avoidance, and obstacle detection in a cost-effective, compact, lightweight, and computationally efficient manner while utilizing limited resources. The MSCKF algorithm typically tracks the uniformly distributed FAST [16] features in the environment using either optical flow [17,18] or descriptors [19] and performs measurement updates when the tracked features are either lost in the current frame (i.e., could not be tracked) or when the IMU/camera pose corresponding to the observed feature is marginalized from the state vector.

Once the MSCKF update is performed and the IMU/camera poses in the state vector are corrected, the MSCKF can triangulate the tracked features in real time, provided they have been observed more than once and the tracking accuracy is reliable. However, applications such as path planning, collision avoidance, and obstacle detection require a dense or sparsely dense depth map of the environment. Traditional optical flow and descriptor techniques are susceptible to noise and changing lighting conditions. Furthermore, feature points cannot be extracted from textureless surfaces. While the MSCKF can triangulate points (not just features) in real time, the lack of features in the environment (such as textureless surfaces like walls and the sky), the limitation of optical flow and descriptor tracking methods to features only, and the limitation of computational resources like GPUs on smaller agile platforms restrict the application of the MSCKF to localization only. Although several efforts have been made recently to address the highlighted issues, such as Zibin Wu’s proposed feature point mismatch removal method [20], which complements RANSAC, Chao Feng Yuan proposed a combined point and line SLAM in dynamic environments [21], Cong Zhang proposed a monocular depth estimation network [22], and Jiale Liu proposed enhanced visual SLAM leveraging point-line features [23]. However, none of these methods addressed the following issues:Limitations of feature point extraction and tracking techniques in environments with limited features;Depth map without estimating features and featureless pixel points in the state vector, given limited computational resources.

To address these problems, we propose a visual–inertial odometry system based on pixel point tracking using reference features from the OpenVINS framework, which simultaneously performs localization and depth estimation of the environment. The main contributions are as follows:To achieve a dense or sparsely dense depth map of the environment, where feature points could not be extracted across the entire image, especially in regions such as walls or planes, an adaptive reference features-based tracking algorithm is proposed. This algorithm can track featureless pixel points across sequential images. The depth of the tracked pixel points is estimated using the MSCKF triangulation method;The traditional VIO tracking methods rely on feature points based on optical flow or descriptors, both of which are susceptible to lighting conditions and camera motion. When feature points cannot be tracked using either of these techniques, such as optical flow or descriptors, even if they remain visible in the image, our pixel point tracking method, which utilizes reference features, can still track any pixel point by using the reference features in the broader vicinity.

After applying the proposed technique, we can estimate the depth of nearly all pixels in the image, except in situations like the sky, where no reference features are available in the vicinity. The remainder of this article, in Section 2, outlines the architecture of the proposed algorithm. Section 3 provides a detailed description of our grid-based pixel point extraction method. Section 4 elaborates on our pixel point tracking algorithm. Section 5 describes the depth estimation and representation methodology. Section 6 presents the experimental results using open-source visual–inertial datasets supported by OpenVINS. Finally, Section 7 concludes and discusses future work.

## 2. System Overview

The method proposed in this paper is mainly an improvement based on the OpenVINS framework [14]. The designed system block diagram is shown in Figure 1, which is primarily divided into the MSCKF-VIO framework, grid-based pixel point extraction, tracking, and triangulation for depth map estimation. The MSCKF-VIO framework employs the same strategy as OpenVINS, mainly including initialization, state propagation, and MSCKF measurement updates. The FAST feature tracker in OpenVINS is initialized by extracting FAST features through a grid-based methodology to ensure a minimum pixel distance between the features. These features are tracked using either pyramidal LK optical flow [24] or ORB descriptors [25]. After being observed more than once, the tracked features can be triangulated using the corresponding camera poses. These tracked features are then sorted according to the number of tracks. The features observed from all camera poses in the state vector are treated as SLAM features and are estimated in the state vector alongside the IMU poses and calibration parameters. The remaining features are treated as MSCKF features, which are only used to correct the IMU pose after triangulation and are not estimated in the state vector.

Our tracking algorithm considers the MSCKF and SLAM features as reference features. After estimating the sliding window of IMU/camera poses from the FAST features (i.e., MSCKF and SLAM features), pixel points are extracted in a grid-based structure, similar to OpenVINS. It is ensured that every grid contains at least one pixel point, which can be a feature point, a point on the contour (i.e., contour points), or a featureless pixel point that is arbitrarily selected as the grid’s center. The grids are kept smaller (5 × 5 pixels) near the center of the image to create a dense depth map, while larger grids are used in other areas. The grid size determines the number of grids in the image, dictating the density of pixel points used to represent the depth map.

The pixel points are tracked using a pyramidal implementation of LK optical flow. The pixel points that are not successfully tracked with optical flow are then tracked using reference features. Finally, Random Sample Consensus (RANSAC) [26,27,28] is applied to remove any outliers. After successful tracking, each tracked point is assigned an ID similar to that used in OpenVINS, and the feature database is updated. Pixel point triangulation is performed using measurements from consecutive images and the corresponding camera poses, thus resulting in a real-time depth map of the image.

The MSCKF-based VINS framework, shown in Figure 1, has already been implemented in several open-source systems, such as MSCKF-VIO [29] and OpenVINS [30], which are publicly available online. This research is developed using the existing framework of OpenVINS, an open platform based on MSCKF and Extended Kalman Filtering, developed by the Robot Perception and Navigation Group (RPNG) laboratory. OpenVINS offers modular management of state vectors and feature databases to facilitate user expansion. A detailed description of the methods and algorithms of the OpenVINS framework can be found in [31]. The MSCKF-VIO framework’s output includes the estimated sliding window of N camera poses and reference features.

Our significant contribution in this paper is the design of a tracking algorithm that uses reference features to track both features and featureless pixel points across sequential images. This approach enables depth estimation for the entire image, except in regions where reference features are unavailable in the vicinity, such as the sky.

## 3. Gridded Pixel Point Extraction Algorithm

The extraction of pixel points is the first phase in estimating the depth of gridded sparse pixel points in the image. As the camera moves, some pixel points move out of the image, and new pixel points must be extracted from the newly observed space. Therefore, the extraction and initialization of pixel points is a continuous process in VIO. Furthermore, an image consists of tens of thousands of pixels, and depth estimation for all pixel points is extremely computationally expensive in real time, especially with limited computational resources, such as in the absence of GPUs.

The image is divided into several grids to restrict the computational load, with one pixel point selected in each grid. Usually, the direction of motion is toward the center of the image. Thus, the density of extracted pixel points is kept higher around the center of the image and lower outside this region. Therefore, the size of the grids around the image center is kept smaller, while it is bigger elsewhere. Grid-based extraction also ensures a uniform distribution of pixel points in the depth map of the image. Since the required density of pixel points differs in the two image regions, we define two distinct areas within each image and utilize two different trackers for this purpose. The number of feature points alone is insufficient to meet the constraint of one pixel point per grid in the occupancy grid map. Therefore, other methods to extract pixel points are considered.

The contours in the image are extracted as a continuum of pixel points, all of which exhibit similar properties, preventing us from characterizing these pixels as features. These points on the contours can also serve as candidate pixel points to be tracked when there is no FAST feature in a grid. Grids containing neither features nor contours can be categorized as textureless planes. All the pixels in these grids exhibit similar properties, and any pixel could be chosen as a candidate pixel point for the depth map.

In the first stage, FAST corner point features above a certain threshold are extracted from the two regions using the FAST [16] corner detection algorithm from OpenCV [32]. The FAST features are refined and sorted according to the maximum response value, from which the strongest FAST feature is selected. Each grid is scanned for these FAST features. If one or more FAST features are present in a grid, the one with the maximum response is selected as a candidate point for tracking, and the occupancy grid map is updated. Since it is extremely unlikely to extract FAST feature pixel points in each grid, as shown in Figure 2a, given that the grid size is very small, i.e., in the range of 5 to 10 pixels in both directions, the points on the contours serve as a convenient choice for the unoccupied grids in the occupancy grid map.

To extract contours, the image is convolved with a Gaussian kernel for blurring, the Canny edge detector is applied to detect the edges, binary image thresholding is performed to obtain a bi-level image, and lastly, OpenCV is used to extract the coordinates of the pixel points on the edges of the bi-level image, as shown in Figure 2b. These contour points are then scanned for each unoccupied grid in the occupancy grid map. Next, we update our tracker by choosing at least one contour point for the unoccupied grids that contain a contour point.

For grids without any features or contours, i.e., textureless or noisy patches in the image, any pixel can be taken as a candidate point to be tracked, i.e., arbitrary points. To ensure uniformity of pixel distribution in the depth map, we select the center of the grid as an arbitrary point to be tracked and initialized in the tracker. Figure 2c shows the extraction of pixel points around the image center with a grid size of 5 × 5 pixels from the image taken from the EuRoC dataset V1_01_easy, showing the three different types of pixel points and the count of the different kinds of pixel points extracted around the image center. Similarly, Figure 2d shows the extraction of pixel points outside the ROI (region of interest) defined around the image center with a grid size of 8 × 8 pixels, showing the three different types of pixel points and their counts. As mentioned earlier, the density requirement for pixel points far from the image center is lower.

In Figure 2, neither the FAST features nor the contour points can be found in several grids corresponding to textureless surfaces, such as carpeted floors, windows, or walls. Consequently, several pixel points (1855 points around the center and 1941 points outside the ROI around the center) are assigned as arbitrary points. The pixel point extraction process is illustrated as a flowchart in Figure 3. Additionally, the number of pixel points (pixel density) chosen within the ROI around the center is significantly higher than in other areas, even though it is four times larger. The reason is apparent: the grid size defines the pixel density.

## 4. An Adaptive Threshold-Based Pixel Point Tracking Algorithm Using Reference Features

To estimate the depth of pixel points in the image, both the newly extracted pixel points and those tracked from the previous image must be tracked with sufficient accuracy, allowing for triangulation for depth estimation. For this purpose, the pyramidal LK optical flow is a convenient choice. Since the optical flow is calculated for pixel points with distinct characteristics, namely features, many points, including the three types of points, i.e., features, contour points, and arbitrary featureless pixel points, cannot be tracked using optical flow.

To determine the corresponding positions of untracked pixel points in the image, the feature points from the VIO framework, whose depths have been estimated in adjacent images, serve as reference features. The invariance of the relative positions between the points to be matched and the reference feature points in adjacent image frames is utilized to estimate the matching position of the pixel points in subsequent image frames. However, due to the differing observation positions of points in space, the relative positions of the points in various frames will change. To effectively match featureless points in adjacent frames, selecting a reference position relative to the point to be estimated that does not change with camera motion is essential.

To address this problem, we propose an adaptive threshold-based reference feature selection algorithm for tracking featureless pixel points, which adaptively selects the set of reference features for a given point while satisfying the aforementioned constraint. Once the set of reference features is known, the depth range of a given point can be calculated using the reference features in the neighborhood. Given the reference features that satisfy the positional invariance constraint concerning the point to be tracked and the depth range around that point, we can track the position of the pixel point in the image using the geometry of the selected reference features in the two images. After tracking all the given pixel points, the 8-point RANSAC outlier rejection algorithm is applied to remove any outliers. The detailed tracking process is explained below.

### 4.1. An Adaptive Threshold-Based Reference Feature Selection Algorithm for Tracking Featureless Pixel Points

Consider the projected coordinates of a pixel point, $P_{1}^{0}$, in an image, 0, with depth $z_{1}$ and the camera intrinsics matrix, K, as follows:(1)$$\left[\begin{array}{c} u_{1}^{0}\\ v_{1}^{0}\\ 1 \end{array}\right]=\frac{1}{z_{1}}KP_{1}^{0}$$
where $\left(u_{1}^{0},v_{1}^{0}\right)$ represent the projected coordinates of a pixel point, $P_{1}^{0}$. The distance between the two points, $P_{1}^{0}$ (point to be matched) and $P_{2}^{0}$ (reference point), in image 0, without considering rotation and translation, is given as follows:(2)$$\left[\begin{array}{c} u_{1}^{0}\\ v_{1}^{0}\\ 1 \end{array}\right]-\left[\begin{array}{c} u_{2}^{0}\\ v_{2}^{0}\\ 1 \end{array}\right]=K\left(\frac{P_{1}^{0}}{z_{1}^{0}}-\frac{P_{2}^{0}}{z_{2}^{0}}\right)$$

After the camera pose changes, the position of $P_{1}^{0}$ in the image plane is given as follows:(3)$$\left[\begin{array}{c} u_{1}^{rt}\\ v_{1}^{rt}\\ 1 \end{array}\right]=\frac{1}{z_{1}^{rt}}KR\left(P_{1}^{0}-t\right)$$
where $z_{1}^{rt}$ is the depth of $P_{1}^{0}$ after the camera pose changes. By solving Equation (3) after camera displacement and rotation, the image plane coordinates are obtained as follows:(4)$$\begin{array}{c} \left[\begin{array}{c} u_{1}^{rt}\\ v_{1}^{rt}\\ 1 \end{array}\right]=\frac{1}{z_{1}^{rt}}K\left[\begin{array}{ccc} r_{11} & r_{12} & r_{13}\\ r_{21} & r_{22} & r_{23}\\ r_{31} & r_{32} & r_{33} \end{array}\right]\left[\begin{array}{c} x_{1}^{0}-t_{x}\\ y_{1}^{0}-t_{y}\\ z_{1}^{0}-t_{z} \end{array}\right]\\ \left[\begin{array}{c} u_{1}^{rt}\\ v_{1}^{rt}\\ 1 \end{array}\right]=\frac{1}{z_{1}^{rt}}K\left[\begin{array}{c} r_{11}\left(x_{1}^{0}-t_{x}\right)+r_{12}\left(y_{1}^{0}-t_{y}\right)+r_{13}\left(z_{1}^{0}-t_{z}\right)\\ r_{21}\left(x_{1}^{0}-t_{x}\right)+r_{22}\left(y_{1}^{0}-t_{y}\right)+r_{23}\left(z_{1}^{0}-t_{z}\right)\\ r_{31}\left(x_{1}^{0}-t_{x}\right)+r_{32}\left(y_{1}^{0}-t_{y}\right)+r_{33}\left(z_{1}^{0}-t_{z}\right) \end{array}\right] \end{array}$$

Therefore, the corresponding depths of points $P_{1}^{0}$ and $P_{2}^{0}$ after camera rotation and translation are given as follows:(5)$$\begin{array}{c} z_{1}^{rt}=r_{31}\left(x_{1}^{0}-t_{x}\right)+r_{32}\left(y_{1}^{0}-t_{y}\right)+r_{33}\left(z_{1}^{0}-t_{z}\right)\\ z_{2}^{rt}=r_{31}\left(x_{2}^{0}-t_{x}\right)+r_{32}\left(y_{2}^{0}-t_{y}\right)+r_{33}\left(z_{2}^{0}-t_{z}\right) \end{array}$$

After the camera pose changes, the distance between the two points in space in the image plane is given as follows:(6)$$\begin{array}{c} \left[\begin{array}{c} u_{1}^{rt}\\ v_{1}^{rt}\\ 1 \end{array}\right]-\left[\begin{array}{c} u_{2}^{rt}\\ v_{2}^{rt}\\ 1 \end{array}\right]=\frac{1}{z_{1}^{rt}}\mathrm{KR}\left(P_{1}^{0}-t\right)-\frac{1}{z_{2}^{rt}}\mathrm{KR}\left(P_{2}^{0}-t\right)\\ \left[\begin{array}{c} u_{1}^{rt}\\ v_{1}^{rt}\\ 1 \end{array}\right]-\left[\begin{array}{c} u_{2}^{rt}\\ v_{2}^{rt}\\ 1 \end{array}\right]=\mathrm{KR}\left(\frac{P_{1}^{0}}{z_{1}^{rt}}-\frac{P_{2}^{0}}{z_{2}^{rt}}\right)-\mathrm{KR}\left(\frac{t}{z_{1}^{rt}}-\frac{t}{z_{2}^{rt}}\right) \end{array}$$

The first part, i.e., $KR\left(\frac{P_{1}^{0}}{z_{1}^{rt}}-\frac{P_{2}^{0}}{z_{2}^{rt}}\right)$, refers to the distance within the image plane without the effect of translation, while the second part, $KR\left(\frac{t}{z_{1}^{rt}}-\frac{t}{z_{2}^{rt}}\right)$, refers to the effect of translation on the distance within the image plane. The following conclusions can be drawn from the analysis above:Camera rotation will cause changes in the relative position of points in space on the imaging plane;Camera displacement will cause changes in the relative position of points in space on the imaging plane.

The interval between consecutive frames is small, i.e., R≈I, then $z_{1}^{rt}\approx z_{1}^{0}$, $z_{2}^{rt}\approx z_{2}^{0}$. Therefore, the effect of camera rotation on depth can be ignored. If the camera displacement is smaller relative to the depth, we can formulate a constraint equation for selecting reference features as follows:(7)$$\frac{\left\|\frac{t}{z_{1}^{rt}}-\frac{t}{z_{2}^{rt}}\right\|}{\left\|\frac{P_{1}^{0}}{z_{1}^{rt}}-\frac{P_{2}^{0}}{z_{2}^{rt}}\right\|}<T_{t}$$

Hence, it can be considered that changes in the relative positional relationship between spatial points in the space within the imaging plane can be ignored if the ratio, $T_{t}$, is less than a certain threshold. If the reference feature, $P_{2}^{0}$, and the point to be matched, $P_{1}^{0}$, satisfy this constraint, we can select $P_{2}^{0}$ as a reference feature with sufficient invariance that can be used to estimate the position of $P_{1}^{0}$ in the adjacent frame, i.e., $P_{1}^{1}$.

### 4.2. Depth Range Estimation

If we can select at least three reference features that form a triangle in both images, and the interior angle values of the triangles in the imaging planes remain unchanged at different times after the camera pose changes (i.e., within a threshold), we can approximately conclude that the change in the camera pose does not affect the relative relationship between the reference features and the pixel point to be matched in the imaging plane.

In practical applications, the depth of the point to be estimated, i.e., both $z_{1}^{0}$ and $z_{1}^{rt}$, is unknown. Therefore, the reference feature points around the point to be estimated are used to define the depth range. In cases in which reference features are unavailable in the broader neighborhood, a pixel point may not be successfully tracked, which can occur situations such as the sky, where no reference depths are available in the vicinity. The translation in Equations (3)–(7) is calculated using the estimated sliding window of IMU/Camera poses from the VIO framework.

Given the normalized coordinates of $P_{1}^{0}$, i.e., $z_{n}=(u_{n},v_{n})$, and the depths of reference features around $P_{1}^{0}$, the minimum and maximum depths in the neighborhood around $P_{1}^{0}$ are known, i.e., (zCmin,zCmax). Given the depth range around $P_{1}^{0}$, the position of $P_{1}^{0}$ corresponding to the minimum and maximum depths (zCmin,zCmax) in the camera coordinate frame can be calculated as follows:(8)PC10=unvn1zC

Using the same reference features, we can find the minimum and maximum depths around $P_{1}^{0}$ in the next frame, i.e., $\left(z_{1_{min}}^{rt},z_{1_{max}}^{rt}\right)$. However, we still do not know the position of $P_{1}^{0}$ in the next frame, i.e., $P_{1}^{1}$.

### 4.3. Threshold-Based Pixel Point Tracking Algorithm Based on Three Reference Points

After establishing the constraint in Equation (7) for selecting reference features in the adjacent images for a given point to be matched, sets of three reference features are selected individually in the adjacent image frames using the feature ID, forming a triangle, as shown in Figure 4. It is essential to impose the following geometrical constraints, in addition to Equation (7), on the triangles formed by the three reference features in the adjacent images:First, the lengths and interior angles of the triangles should be greater than a certain threshold, such as 10 pixels and 15 degrees;To ensure the similarity of the triangles, the difference in the ratios of the three lengths of the two triangles should be less than a certain threshold, which depends on the accuracy requirement of the depth map. A lower threshold value, such as 0.1, indicates higher similarity than any value greater than 0.1;Given the translation, **t**, depth of the reference feature, $z_{2}^{rt}$, after the camera movement, minimum depth, $z_{1_{min}}^{rt}$, around the point, $P_{1}^{0}$, in image 1, i.e., $P_{1}^{1}$, after camera movement, the position of the reference feature in image 0, i.e., $P_{2}^{0}$, and the position of the point to be tracked in the camera frame in image 0, i.e., PC1min0, calculated with minimum depth zminC, we can apply the constraint Equation (7) to find the ratio, tz1minrt−tz2rtPC1min0z1minrt−P20z2rt;Again, calculate the ratio, tz1maxrt−tz2rtPC1max0z1maxrt−P20z2rt, using the maximum depth, $z_{1_{max}}^{rt}$, around the point, $P_{1}^{0}$, in image 1, i.e., $P_{1}^{1}$ after camera movement and the position of the point to be tracked in the camera frame in image 0, i.e., PC1max0, calculated with the maximum depth, zCmax;Apply the same process to the remaining two reference features and find the threshold ratios. If the six ratios are within the pre-defined threshold limit, $T_{t}$, such as 0.1 or 0.05, the three reference features can be used to find the matching position of $P_{1}^{0}$.

Given the positions of the three reference features in the two images and the position of $P_{1}^{0}$ in image 0, the distance from the point, $P_{1}^{0}$, in the first frame to each side of the virtual triangle can be used to establish the constraints on its position, as shown in Figure 5.

**Figure 4 sensors-25-02849-f004:**
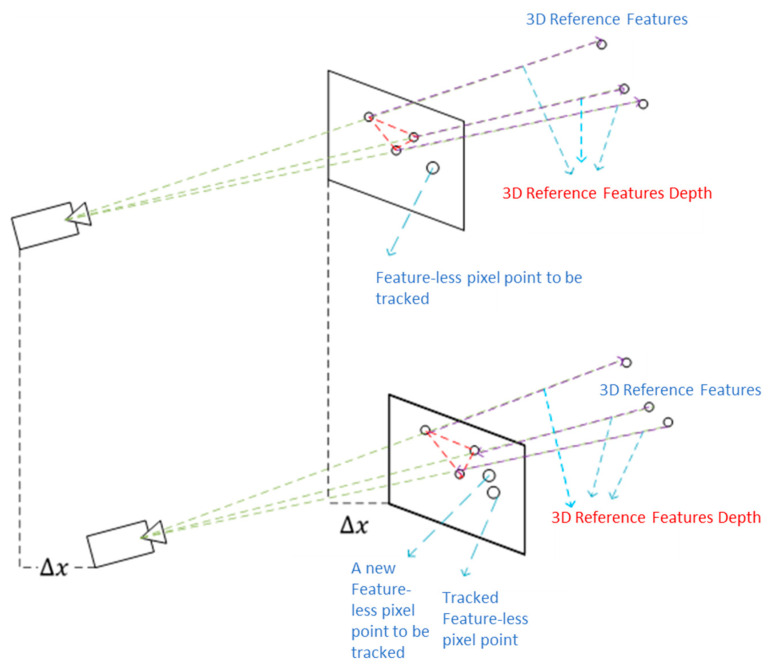
Featureless pixel point tracking algorithm using reference features.

Firstly, the equations of the straight lines satisfying the three sides a,b, and c of the feature triangle in the previous frame are calculated. For example, knowing the two reference feature points on the edge a, the general formula Ax+By+C=0 for the straight line on which a is located can be calculated as follows:(9)$$\begin{array}{c} A=y_{2}-y_{1}\\ \;\;B=-\left(x_{2}-x_{1}\right)\\ C=-A*\;x_{2}-B\;*\;y_{2}=-A\;*\;x_{1}-B\;*\;y_{1} \end{array}$$
where $\left(x_{1},y_{1}\right)$ and $\left(x_{2},y_{2}\right)$ are the two-dimensional image coordinates of the two reference features. Then, calculate the distances, $D_{1}-D_{3}$, from the arbitrary pixel point to be tracked (from the last image) to the three lines of the triangle from the previous image, respectively, as follows:(10)$$D_{n}=\frac{\left|Ax_{0}+By_{0}+C\right|}{\sqrt{A^{2}+B^{2}}}$$
where $\left(x_{0},y_{0}\right)$ is the position of the arbitrary point in the previous frame. These distances, $D_{1}-D_{3}$, define other equations of lines that are perpendicular to the three sides, a,b, and c, of the feature triangle. The lines perpendicular to the three sides, a,b, and c, of the feature triangle can be calculated as follows:(11)$$\begin{array}{c} Ax_{0}+By_{0}+C_{new}=0\;\\ C_{new}=-A\;*\;x_{0}-B\;*\;y_{0} \end{array}$$

Let us denote the perpendicular lines given by Equation (11) as $d_{1},d_{2},d_{3}$. The intersection of lines $d_{1},d_{2},d_{3}$ is the arbitrary feature point, represented by the coordinates (x_0_, y_0_). Similar to finding the equations of the lines of the triangle formed by the three selected reference features in the last frame, calculate the parameters $\left(A_{i},B_{i},C_{i}\right)$ of the three lines in the new frame using the reference feature’s position in the new image frame, as given by Equation (9). Calculate the parameter, $C_{new}$, for the perpendicular lines in the new image using the distances, $D_{1}-D_{3}$, from the arbitrary feature (from the last image) to the three lines of the triangle from the previous image as follows:(12)$$C_{new}=D\cdot \sqrt{{(A}^{2}+B^{2})}+C$$

Depending on whether $C_{new}>C$ in the last frame or $C_{new}<C$, the sign of D in the new frame is positive or negative. Finally, the intersection of any two out of three of these perpendicular lines is the predicted position of the arbitrary feature, as shown in Figure 5. Let us say that the two perpendicular lines are given by $A_{3},B_{3},C_{3_{new}}$ and $A_{4},B_{4},C_{4_{new}}$; the predicted position of the pixel point in the new image frame, $P_{1}^{1}$, can be calculated as follows:(13)$$\begin{array}{c} X=\frac{B_{3}C_{4_{new}}-B_{4}C_{3_{new}}}{A_{3}B_{4}-A_{4}B_{3}}\\ Y=\frac{A_{4}C_{3_{new}}-A_{3}C_{4_{new}}}{A_{3}B_{4}-A_{4}B_{3}}\; \end{array}$$

Depending on the computational resources and the required accuracy, $N_{RF}$, multiple sets of reference features satisfying the above constraints can be used to track the pixel point $P_{1}^{0}$. The estimate can be further refined by taking the arithmetic mean of all the estimates, as shown in Equation (14).(14)$$\begin{array}{c} X=\frac{\sum X_{i}}{N_{RF}}\\ Y=\frac{\sum Y_{i}}{N_{RF}}\; \end{array}$$

### 4.4. RANSAC Outlier Rejection Algorithm

Once all the pixel points are tracked using either optical flow or reference features, only the pixel points tracked within the image are considered, and the eight-point RANSAC [27,28] outlier rejection is applied based on the fundamental matrix derived from the corresponding points in two images. The main idea is that the matching points in the two images should satisfy the following constraint:(15)$${P^{1}}^{T}\cdot [t\times ]\cdot R\cdot P^{0}=0$$
where $P^{0}$ and $P^{1}$ are the homogeneous coordinates of the pixel points in the two image frames, [t×] is the antisymmetric matrix of the displacement vector t, and R is the rotation matrix of the two frames. Due to matching errors, not all matching points can satisfy the abovementioned constraints.

Finally, pixel point extraction is performed in the newly observed or empty image grids, according to Section 3, and the database is updated based on the tracked and newly observed pixel points.

## 5. Depth Estimation

Once the gridded pixel points, comprising feature points, contour points, and featureless pixel points representing the environment, are tracked using either optical flow or the proposed method, the depth of the tracked pixel points is estimated by the MSCKF feature point triangulation algorithm, which is further refined by 3D inverse non-linear optimization, as explained below.

### 5.1. The MSCKF Feature Point Triangulation Algorithm

The MSCKF algorithm maintains a sliding window of N camera poses corresponding to each observed image. The observed pixel point measurements, undistorted normalized image coordinates, and timestamps are stored in the tracker database using a unique ID for each feature. When a certain point is observed in image space more than once, the depth information of the pixel points can be obtained through triangulation in real time, given the corresponding camera poses, as shown in Figure 6.

We take all the poses from which the point is seen to be of known quantity. The pixel points are triangulated in the anchor camera frame {A}, which is the current camera pose. If the pixel point $p_{f}$ is observed by camera poses 1…m, given the anchor pose A, we can derive the following transformation from any camera pose $C_{i},i=1\ldots m$, as follows:(16)pfCi=RACi(pfA−pCiA)pfA=RATCipfCi+pCiA
where pfCi represents the position of the point in the ith camera frame, RACi represents the rotation matrix from anchor frame {A} to the ith camera frame, pfA represents the point position in the anchor frame, which is the current camera frame, and pCiA is the position of the ith camera frame in the anchor frame. In the absence of noise, the measurement in the current frame is the bearing bCi and depth zCi. Thus, we have the following mapping to a point seen from the current frame:(17)pfCi=zfCibfCi=zfCiunvn1
where $u_{n}$ and $v_{n}$ represent the undistorted normalized image coordinates, and zCi is the feature depth in the ith camera frame. This bearing can be warped into the anchor frame by substituting it into Equation (15).(18)pfA=RATCizfCibfCi+pCiApfA=zfCibCi→fA+pCiA

To remove the need to estimate the extra degree of freedom of depth, zfCi, we define the vectors, NiA, orthogonal to the bearing, bCi→fA, and multiply them with Equation (18). By stacking all the measurements, we can obtain the following equation:(19)⋮NiA⋮pfA=⋮NiApCiA⋮

Since each pixel measurement provides two constraints, we will have enough constraints to triangulate the feature. In practice, the more views of the point, the better the triangulation; thus, we usually want the point to be seen from at least three views. To create a square system, we can perform the following “trick”:(20)ATApfA=ATb(∑NiTANiA)pfA=(∑NiTANiApCiA)

This is a 3 × 3 system that can be quickly solved compared to the original 3m×3m or 2m×2m systems. We also check that the triangulated feature is “valid” in front of the camera and not too far away.

### 5.2. Three-Dimensional Inverse Non-Linear Optimization

After obtaining the 3D position of a pixel point, a nonlinear least-squares optimization is performed to refine this estimate. We use the inverse depth representation to achieve good numerical stability, which helps with convergence. The position of the point in $i^{th}$ camera frame can be written as follows:(21)pfCi=RACi(pfA−pCiA)

By substituting the 3D coordinates of the point position in anchor frame {A} into Equation (21), we obtain the following equation:(22)pfCi=RACi(xfAyfAzfA−pCiA)

By dividing Equation (22) by the depth of the point position in the anchor frame zfA, we obtain the following equation:(23)1zfApfCi=RACixfAzfAyfAzfA1−1zfApCiA

By substituting uA=xfAzfA, vA=yfAzfA and ρA=1zfA, we can summarize Equation (23) as follows:(24)huA,vA,ρA=h1h2h3=RACiuAvA1−ρApCiA

Equations (21)–(24) represent the pixel point transformation, where the pixel point position in $i^{th}$ camera frame is multiplied by the inverse depth zfA. Finally, the normalized point measurement seen from the “ith” camera frame, $C_{i}$, can be written as follows:(25)$$\hat{z}=H(u_{A},v_{A},\rho_{A})=\left[\begin{array}{c} u_{i}\\ v_{i} \end{array}\right]=\left[\begin{array}{c} \frac{h_{1}}{h_{3}}\\ \frac{h_{2}}{h_{3}} \end{array}\right]$$

The least squares can now be formulated as follows:(26)|uA,vA,ρAargmin|z−z^||2

The Jacobian of Equation (25) with respect to the parameters $u_{A},v_{A},\rho_{A}$ can be calculated using the chain rule as follows:(27)$$\frac{\delta H}{\delta [u_{A},v_{A},\rho_{A}]}=\frac{\delta H}{\delta h(u_{A},v_{A},\rho_{A})}\frac{\delta h(u_{A},v_{A},\rho_{A})}{\delta [u_{A},v_{A},\rho_{A}]}$$

The Jacobian of Equation (24) with respect to the parameters $\left(u_{A},v_{A},\rho_{A}\right)$ can simply be calculated as follows:(28)δh(uA,vA,ρA)δ[uA,vA,ρA]=RRCi100100−pCiA

The Jacobian of Equation (25) with respect to Equation (24) can be calculated as follows:(29)$$\frac{\delta H}{\delta h(u_{A},v_{A},\rho_{A})}=\left[\begin{array}{c} \begin{array}{ccc} \frac{1}{h_{3}} & 0 & \frac{-h_{1}}{h_{3}^{2}} \end{array}\\ \begin{array}{ccc} 0 & \frac{1}{h_{3}} & \frac{-h_{2}}{h_{3}^{2}} \end{array} \end{array}\right]$$

Finally, the Jacobian $\frac{\delta H}{\delta [u_{A},v_{A},\rho_{A}]}$ can be calculated as follows:(30)δHδ[uA,vA,ρA]=1h30−h1h3201h3−h2h32RACi100100−pCiA=1h30−h1h3201h3−h2h32RACi100100pACi

Since RACipCiA=−pACi and pACi=XACiYACiZACi, when solving for RACi=R0,0R0,1R0,2R1,0R1,1R1,2R2,0R2,1R2,2, Equation (30) can also be written as follows:(31)δHδuA,vA,ρA=1h30−h1h3201h3−h2h32R0,0R0,1XACiR1,0R1,1YACiR2,0R2,1ZACi=R0,0h3−R2,0h1h32R0,1h3−R2,1h1h32XACih3−h1ZACih32R1,0h3−R2,0h2h32R1,1h3−R2,1h2h32YACih3−h2ZACih32

Once the Jacobians are formulated, the Gauss–Newton or Levenberg–Marquardt optimization can be performed as an iterative process until convergence.

### 5.3. Gridded Depth Representation of an Image

After successfully tracking and triangulating pixel points spaced 5 to 10 pixels apart throughout the image, the image depth can be represented by dividing it into several grids, each containing multiple triangulated pixel points. The depth information from these pixel points can represent the minimum, maximum, and average depth of each grid.

## 6. Experimental Results and Discussion

To verify the effectiveness of the proposed method based on pixel point tracking using reference features, experiments are conducted in an indoor environment using the publicly available EuRoC visual–inertial dataset [15]. This approach is instrumental when featureless pixel points and feature points cannot be tracked using traditional techniques, such as optical flow or descriptors. The EuRoC dataset contains 11 sequences of stereo images, each with a size of 752 × 480 pixels, synchronized IMU information at 200 Hz, ground truth, and calibration information for the sensors’ external and internal parameters.

This research focuses on monocular VIO, considering only one IMU/camera pair. The experiments are conducted on three sequences from the EuRoC dataset. The experimental platform is a Xiaomi Redmi G 2022 laptop purchased from Beijing, China, with an Intel I7-12650H @ 2.3GHz, 16 GB of RAM, no GPU acceleration, and the 64-bit Ubuntu 20.04 WSL operating system.

When matching is performed reasonably accurately, the MSCKF triangulation algorithm can successfully triangulate the tracked points. The greater the number of tracks for a point, the higher the depth estimation accuracy. The depth of every selected tracked pixel point cannot be estimated successfully in real time because of several constraints, such as RANSAC, the availability of reference features in the neighborhood, and the calculation of depth with fewer tracks. For these reasons, the grid size should be selected carefully, as it affects the density of the depth map and the computational cost. To ensure that the depth map is useful for applications such as obstacle detection, collision avoidance, or path planning, the image is divided into several grids, each containing several triangulated pixel points. When there are more than N triangulated pixel points in a grid, we can estimate the average depth of that grid, provided the grid size is small enough to be represented by N pixel points.

### 6.1. V1_01_Easy Sequence

The V1_01_easy sequence comprises 2912 timestamped monocular images with IMU measurements at 200 Hz. Figure 7a shows the depth map at the beginning of the sequence when sufficient measurements are unavailable. Figure 7b shows the average depth of each grid, where less than 50% of the grids could be estimated. This is because reference features are not available around the remaining grids. However, the depth of most of the grids in Figure 7c,d, especially around the center, could be estimated. Moreover, it can also be seen that the highlighted grids in Figure 7 do not have any reference features in the neighborhood, which also restricts the tracking of pixel points in these grids. Moreover, the camera motion in Figure 7 is rightward. Hence, the pixel points in the rightmost grids do not have sufficient tracks for triangulation. However, for all the grids in which reference features are available in the neighborhood, the depth of the grids could be estimated.

It can further be seen in Figure 7 that our reference feature-based tracking algorithm was applied 5520 times to obtain the image in Figure 7c and 11,958 times to obtain the image in Figure 7d, when the optical flow technique was not successful. Furthermore, triangulation is performed based on camera translation; if the translation is too small or the camera motion only consists of pure rotation, triangulation cannot be performed successfully. Moreover, the EuRoC V1_01_easy dataset consists of 2912 image frames, from which we can estimate more than 90% of grids in 736 frames when the image is divided into 15 × 15 grids. Feature extraction around the image center is performed with 5 × 5 grids and 8 × 8 grids elsewhere, with one feature in each grid.

Another experiment was conducted to compare the proposed method with the pyramidal LK optical flow method. First, the extracted FAST features are tracked using pyramidal LK optical flow and triangulated using the MSCKF feature point triangulation method. These data serve as the ground truth. The same FAST features are also tracked and triangulated using the proposed method. It can be seen in Table 1 and Table 2 that the depths calculated from the two methods differ by only a few centimeters. Due to space limitations, the estimated depths of only a few features tracked with both methods are shown in Table 1 and Table 2 for adjacent frames of the EuRoC dataset v1_01_easy.

In another experiment on the v1_01_easy sequence, the pixel points were tracked 30,000 times using the proposed method. The average time to track a pixel point successfully using reference features is 0.001034 s, without using any parallel programming techniques, and the minimum time to track a pixel point is 0.000038523 s.

### 6.2. V1_02_Medium Sequence

A similar experiment was performed on the v1_02_medium sequence, which consisted of 1710 monocular images. It can be seen in Figure 8 that we can estimate the average depth of most of the grids in the entire sequence, especially toward the center of the image. The following constraints were observed:Insufficient translation at the beginning to triangulate the observed points due to the initialization phase of MSCKF-VIO;The unavailability of reference features in the neighborhood of pixel points to be tracked leads to a reduced density of triangulated pixel points;Triangulation is performed when a point is observed for at least five measurements. Thus, points with fewer than five measurements were not triangulated, which reduced the density of triangulated pixel points.

Depending on the accuracy requirements and the density of the depth map, the constraints of five measurements for triangulation or the number of triangulated pixel points necessary for calculating the average depth of the grid can be adjusted. Under the current parameters, i.e., five measurements of three estimated points, we can estimate the depths of most of the grids in each image. It is also important to note that during the sequence of 1710 image frames, the optical flow technique was applied 11,291,373 times, while the proposed technique was successfully applied 174,680 times, proving its usefulness.

As shown in Figure 8, we cannot estimate the depths of around 25% of the grids in several images of this sequence. This is because the MSCKF algorithm is sensitive to camera translation. Both very small and very large translations affect the MSCKF triangulation capabilities. Additionally, the lack of distinct characteristics within the grid or in the vicinity also affects the estimation results.

**Figure 8 sensors-25-02849-f008:**
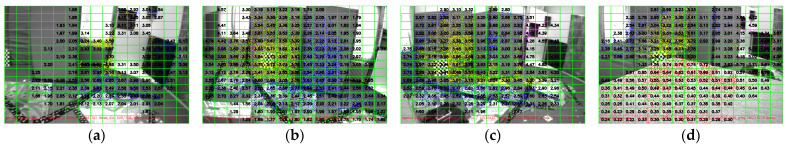
(**a**–**d**) A 15 × 15 gridded depth map of the images at different timestamps from the v1_02_medium sequence of the EuRoC dataset. The pixel points are extracted using a grid size of 5 × 5 pixels around the center and 8 × 8 pixels elsewhere. Triangulation is performed when a pixel point is tracked across five frames. The various colors of the pixel points represent different depth ranges. The average depth of each grid is calculated when the number of triangulated pixels in that grid is three or more.

Similar to Table 1 and Table 2, we tracked the known feature points using two methods, i.e., pyramidal LK optical flow and the proposed tracking method using reference features. Due to space limitations, the estimated depths of the points tracked with the two methods are shown in Table 3 and Table 4 for two random images. This supports our assertion that the proposed method can track pixel points with adequate accuracy based on the reference features in the neighborhood. Furthermore, the depths estimated after tracking with the two methods differ by only a few centimeters.

For the v1_02_medium sequence, the proposed technique was applied 337,004 times and was successful 174,680 times. It was determined that the average time to track a pixel point using reference features successfully is 0.0003769 s, without utilizing any parallel programming techniques, and the minimum time to track a pixel point is 0.0000102 s.

### 6.3. MH_01_Easy Sequence

The MH_01_easy dataset comprises 3682 monocular images. Consistent with our previous findings, the proposed algorithm can track and estimate the depth of most grids in the image that possess distinct characteristics either within or nearby, as illustrated in Figure 9. The algorithm was applied successfully 153,785 times during the sequence when the optical flow method could not track pixel points. The minimum and average times to successfully track a pixel point are 0.00002972 and 0.002932774 s, respectively. As shown in Figure 9, we can estimate over 75% of the grids in most images within this sequence.

### 6.4. Comparison and Timing Details

A detailed comparison of the proposed method for the sequences mentioned above with the generated ground truth is available on Google Drive, as stated in the Data Availability Statement for all three sequences. For the v1_01_easy sequence, each feature ID is accompanied by the depth estimated using pyramidal Optical Flow tracking and the depth estimated by the proposed method. For the other sequences, the feature ID is followed by its timestamp and the estimated depths from both methods. The time taken to track a single pixel, the average time spent tracking pixels up to the specified timestamp, and the total number of tracked pixels are provided, along with videos of each sequence displaying the estimated grid depths.

## 7. Conclusions and Future Work

In this paper, we propose a monocular visual–inertial odometry-based gridded image depth map estimation algorithm. The proposed method can track pixel points when traditional techniques, such as optical flow or descriptors, cannot. These pixel points can be classified as either features or featureless points. The proposed method is built on OpenVINS, and depth estimation is performed using the MSCKF triangulation algorithm. Furthermore, the system relies on a monocular IMU/camera pair, avoiding the need for expensive RGBD cameras or additional computational resources such as GPUs. The proposed method has been tested extensively on three sequences of the EuRoC dataset. Since the proposed method is based on monocular visual–inertial odometry, it is validated on the well-known visual–inertial dataset EuRoC. Based on the analysis and results, three conclusions are drawn as follows:When traditional techniques, such as optical flow or descriptors, cannot track a pixel point that either has distinct characteristics (i.e., a feature) or is a featureless pixel point on a textureless surface, such as walls or glass windows, the proposed method can track these pixel points using reference features in the neighborhood in a gridded, threshold-based adaptive manner;The depth estimation of pixel points utilizes the estimated sliding window of IMU/camera poses from the MSCKF-VIO framework;A gridded depth map of the environment is generated by dividing the image into several grids. The depth of a grid is estimated when at least N pixel points within the grid have had their depths estimated. Finally, the minimum or average depth of the grids can be utilized for applications such as obstacle detection or path planning.

In the future, this paper will improve in several directions, including validating the proposed method on outdoor datasets such as KITTI [33], which is currently unsupported by OpenVINS, likely due to the lower data rate of the IMU and different coordinate frame definitions compared to standard visual–inertial datasets. This also involves maximizing existing computational resources through parallel programming to minimize tracking time, utilizing epipolar geometry to reduce tracking errors, and optimally configuring various thresholds based on different situations.

## Figures and Tables

**Figure 1 sensors-25-02849-f001:**
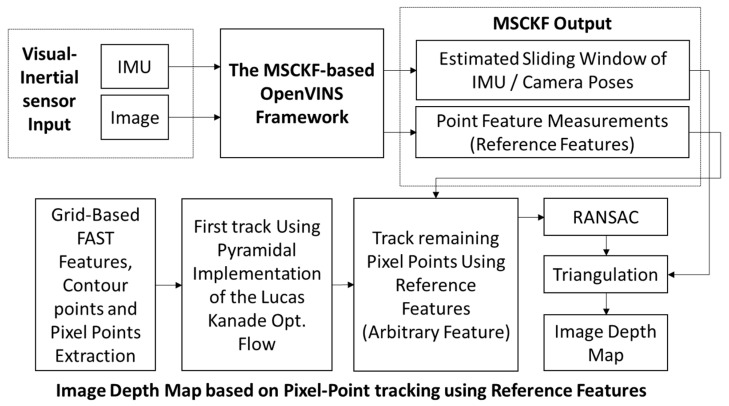
Architecture of the proposed system.

**Figure 2 sensors-25-02849-f002:**
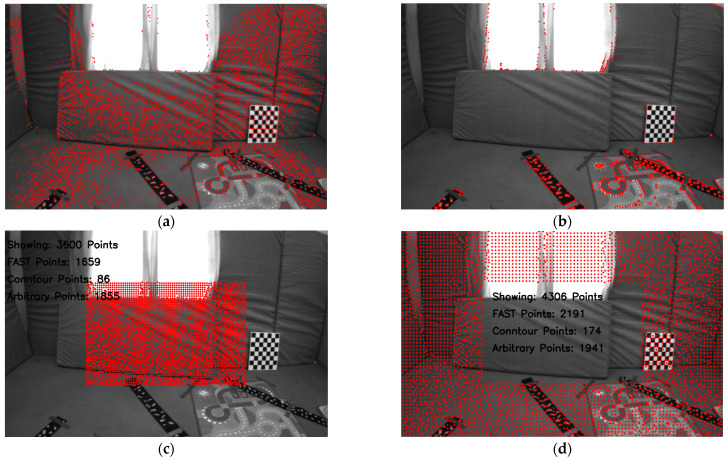
Pixel point extraction method: (**a**) extracted FAST features; (**b**) extracted contour points; (**c**) extracted pixel points surrounding the center of the image; (**d**) pixel points extracted outside the region defined around the center of the image.

**Figure 3 sensors-25-02849-f003:**
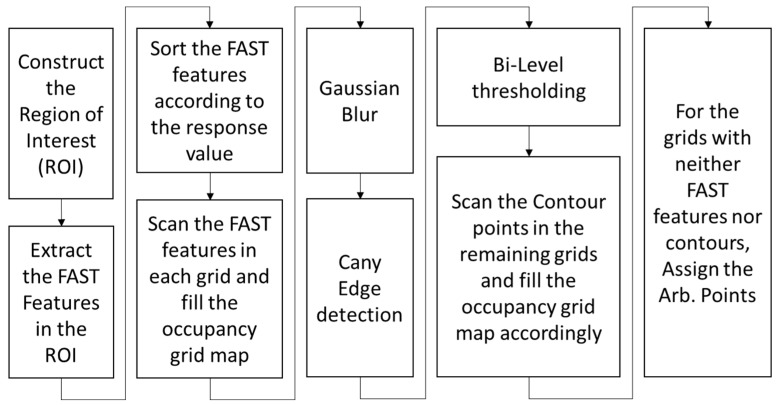
Extraction process of the three different types of pixel points around and outside the image center.

**Figure 5 sensors-25-02849-f005:**
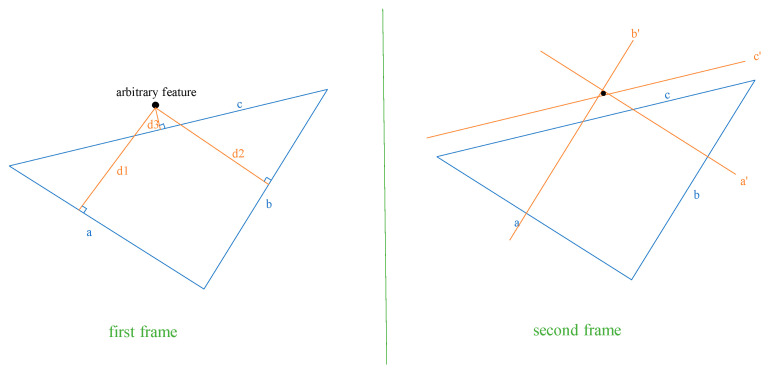
Schematic diagram for tracking arbitrary pixel points.

**Figure 6 sensors-25-02849-f006:**
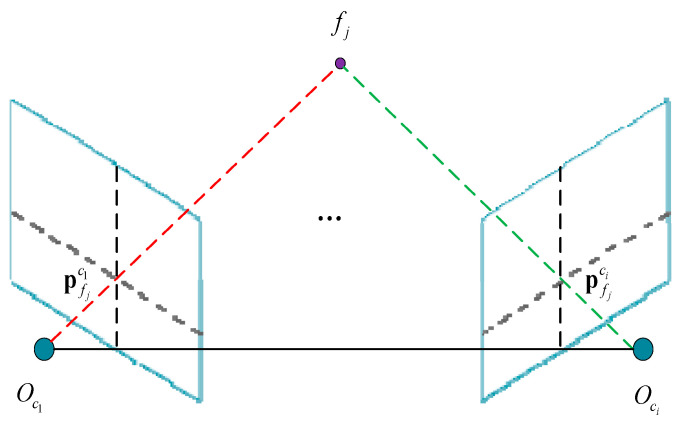
Schematic diagram of triangulation of points.

**Figure 7 sensors-25-02849-f007:**
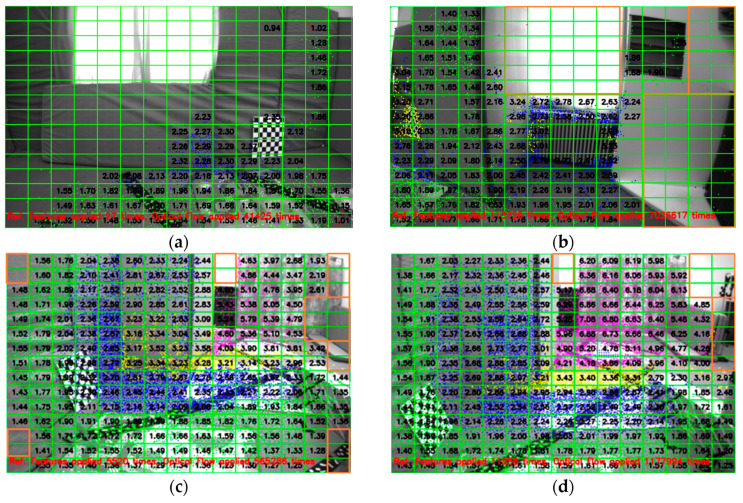
(**a**–**d**) A 15 × 15 gridded depth map of the images shows the average depth of each grid from multiple views of the v1_01_easy sequence of the EuRoC dataset. The pixel points are extracted with a grid size of 5 × 5 pixels around the center and 8 × 8 pixels elsewhere. Triangulation is performed when a pixel point is tracked across five frames. The various colors of the pixel points represent different depth ranges. The average depth of each grid is calculated when the number of triangulated pixels in that grid is five or more.

**Figure 9 sensors-25-02849-f009:**
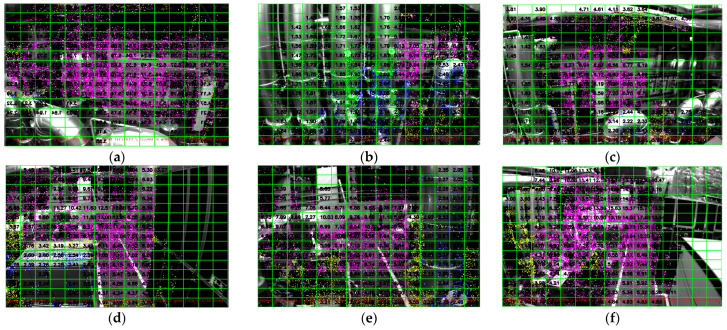
(**a**–**f**) A 15 × 15 gridded depth map of the images at different timestamps from the MH_01_easy sequence of the EuRoC dataset. Pixel points are extracted using a grid size of 5 × 5 pixels around the center and 8 × 8 pixels elsewhere. Triangulation is performed when a pixel point is tracked across five frames. The different colors of the pixel points show different depth ranges. The average depth of each grid is calculated when there are five or more estimated pixels in that grid.

**Table 1 sensors-25-02849-t001:** Comparison of the depth estimates from the two methods (pyramidal LK optical flow and the proposed method) at image timestamp 1403715281.112143039703369.

Feature ID	Estimated Depth Using the Pyramidal Implementation of LK Optical Flow (m)	Estimated Depth Using the Proposed Method (m)
4258	1.509639	1.516483
4259	1.574686	1.570221
4261	1.521715	1.527992
4262	1.449927	1.461694
4263	1.544786	1.556617

**Table 2 sensors-25-02849-t002:** Comparison of the depth estimates from the two methods (pyramidal LK optical flow and the proposed method) at image timestamp 1403715281.162142992019653.

Feature ID	Estimated Depth Using the Pyramidal Implementation of LK Optical Flow (m)	Estimated Depth Using the Proposed Method (m)
4258	1.50875	1.530322
4259	1.570865	1.572788
4261	1.519244	1.537466
4262	1.452665	1.484772
4263	1.544601	1.563351

**Table 3 sensors-25-02849-t003:** Comparison of the depth estimates from the two methods (pyramidal LK optical flow and the proposed method) at image timestamp 1403715598.16214 in the v1_02_medium sequence.

Feature ID	Estimated Depth Using the Pyramidal Implementation of LK Optical Flow (m)	Estimated Depth Using the Proposed Method (m)
46065	2.384801	2.428201
45377	2.534047	2.607109
46132	2.380807	2.399241

**Table 4 sensors-25-02849-t004:** Comparison of depth estimates from the two methods (pyramidal LK optical flow and the proposed method) at image timestamp 1403715598.26214 in the v1_02_medium sequence.

Feature ID	Estimated Depth Using the Pyramidal Implementation of LK Optical Flow (m)	Estimated Depth Using the Proposed Method (m)
46065	2.252892	2.287129
45377	2.39568	2.448431
46132	2.241698	2.258009
46435	1.472409	1.546551

## Data Availability

The original data presented in the study are openly available at https://drive.google.com/drive/folders/1gyFMjZYSkTrxM2poNRtX6fwxXu1DVjcY?usp=drive_link (accessed on 8 March 2025).

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
