# Peer review of "An Adaptive Threshold-Based Pixel Point Tracking Algorithm Using Reference Features Leveraging the Multi-State Constrained Kalman Filter Feature Point Triangulation Technique for Depth Mapping the Environment"

_sensors, 2025, doi:10.3390/s25092849_

Round 1
Reviewer 1 Report
Comments and Suggestions for Authors
This paper presents a method for depth estimation of objetcs in the image. The method is divided into the MSCKF-VIO framework of [13], the featured point extraction and tracking, and the point triangulation.
The contribution is in the pixel point extraction and tracking algorithm, where authors use reference features such as corner [15,25], contour and arbitrary pixels.
- The paper presents a large quantity of grammatical errors and the english style is not suitable. I completely understand the effort and difficulty of writing in a foreign language thus, I encourage the authors to be assisted by an English-speaker.
- The abstract does not describe the work carried out neither explains its application and contribution.
- The description of the method is not clear and it is difficult to follow.
- The image is divided into two grids. The size of the grid of the central area of the image is 5x5 pixels, whereas the size of the grid corresponding to the image region around the central zone is 8x8 pixels.
In line 191 it is written the next: “The parameter of grid-size can be changed depending on the requirement of depth map density.” As there is no more explanation, I understand that the authors are who manually change the size of the grid. Thus the adjetive “adaptive” should be remove from the title “3. Adaptive Gridded Threshold based Pixel-Points Extraction Algorithm”.
- In Figure 2, it is said that the three different types of pixel points are shown, however the image just shows some red colour points in the image. It is impossible to distinguish between FAST corner points, contour points and arbitrary points.
- In line 325, it is said, “1. Rotation of the camera changes the depth of the point in the camera frame.” Understandig the depth of the point as the distance between the point and the camera, this distance doesn’t change.
- In line 371 it is written the next: “The difference of ratios of the three lengths of the two triangles should be less than a certain threshold such as 0.1 ensuring the similarity of triangles;” As there is no more explanation, I understand that the authors are who manually change this threshold. Thus the adjetive “adaptive” should be remove from the title “4.3. Adaptive Threshold based pixel point tracking algorithm based on three Reference Points”.
- The introduction focuses on navigation systems so I assume that the proposed method is for a navigation system of some vehicle not specified. The point is that I don´t understand why the images used for testing the method are images of rooms.
The paper presents a large quantity of grammatical errors and the english style is not suitable.
Author Response
The authors thank the reviewer for the valuable feedback on the submitted manuscript. The responses are submitted below:
Comments 1: The paper presents a large quantity of grammatical errors and the english style is not suitable. I completely understand the effort and difficulty of writing in a foreign language thus, I encourage the authors to be assisted by an English-speaker.
Response 1: Agree. Thank you for pointing out the grammatical mistakes, which have been fixed to the best of our ability. The entire manuscript has been thoroughly reviewed and modified to emphasize this point.
Comments 2: The abstract does not describe the work carried out neither explains its application and contribution.
Response 2: Agree. We have, accordingly, revised the abstract to address the shortcomings.
Comments 3: The description of the method is not clear and it is difficult to follow.
Response 3: Agree. We have revised the manuscript to address the clarity. Particularly, the pixel points tracking and depth estimation section is split into two parts in order to improve the clarity.
Comments 4: The image is divided into two grids. The size of the grid of the central area of the image is 5x5 pixels, whereas the size of the grid corresponding to the image region around the central zone is 8x8 pixels. In line 191 it is written the next: “The parameter of grid-size can be changed depending on the requirement of depth map density.” As there is no more explanation, I understand that the authors are who manually change the size of the grid. Thus the adjetive “adaptive” should be remove from the title “3. Adaptive Gridded Threshold based Pixel-Points Extraction Algorithm”.
Response 4: The grid size determines the number of grids in the image, which subsequently dictates the density of pixel points used to represent the depth map. Agree. We have revised the manuscript to address this point which can be found on line 196-197 of the revised manuscript describing the impact of grid size on the density of depth map.
Comments 5: In Figure 2, it is said that the three different types of pixel points are shown, however the image just shows some red colour points in the image. It is impossible to distinguish between FAST corner points, contour points and arbitrary points.
Response 5: Agree. The figure has been updated to address the comment. The figures showing the FAST features and the contour points are added as Figure 2(a) and Figure 2(b).
Comments 6: In line 325, it is said, “1. Rotation of the camera changes the depth of the point in the camera frame.” Understandig the depth of the point as the distance between the point and the camera, this distance doesn’t change.
Response 6: Agree. The pure rotation does not affect the depth. Hence, the mistakes and unnecessary points have been omitted.
Comments 7: In line 371 it is written the next: “The difference of ratios of the three lengths of the two triangles should be less than a certain threshold such as 0.1 ensuring the similarity of triangles;” As there is no more explanation, I understand that the authors are who manually change this threshold. Thus the adjetive “adaptive” should be remove from the title “4.3. Adaptive Threshold based pixel point tracking algorithm based on three Reference Points”.
Response 7: To ensure the similarity of the triangles, the difference in the ratios of the three lengths of the two triangles should be less than a certain threshold, which depends on the accuracy requirement of the depth map. A lower threshold value, such as 0.1, indicates higher similarity than any value greater than 0.1; Agree. We have revised the statement in lines 377-380 to ensure clarity. Further, lines 304-313 can also clarify the purpose of using the word “Adaptive,” which basically refers to the selection of reference features adaptively for tracking a given point satisfying the constraints.
Comments 8: The introduction focuses on navigation systems so I assume that the proposed method is for a navigation system of some vehicle not specified. The point is that I don´t understand why the images used for testing the method are images of rooms.
Response 8: Section 6 presents the experimental results using open-source visual-inertial datasets supported by OpenVINS. Finally, Section 7 concludes and discusses future work. We agree with this comment and have addressed it in several places in the revised manuscript, such as lines 49, 171, and 652. The proposed method is built on the existing OpenVINS framework, which mainly supports the visual-inertial EuRoC dataset sequences. This is the main reason we chose to test our method on the indoor dataset.
Reviewer 2 Report
Comments and Suggestions for Authors
This manuscript proposes a method to generate a depth map of pixels in an image by tracking the Pixel-Points using Reference Features in the neighborhood and estimating the depth using the MSCKF-VIO feature point triangulation method. Although this manuscript's workload seems sufficient, some issues need further clarification.
- In the flowchart in Figure 1, some modules (EKF Estimator, Initialization, and Zero Velocity Detection) are not connected. Readers cannot determine which stage these modules are used in the flowchart. Some readers may also think this is a flowchart module description.
- Figure 3 needs to be redrawn because the text in the flowchart has been deformed and some text is partially obscured.
- The ellipsis used in Eq.(4) is inappropriate, and the formula must be written entirely. Other formulas have the same problem, and we hope the author can supplement the omitted formulas.
- The use of secondary foot tags is not highly recommended. It would be even better if the variable symbols in the text could be redesigned.
- Comparing Figure 4 and Figure 5, it is suggested that the first letter of the explanatory phrases in these figures should be capitalized consistently.
- In the entire text, the use of capital letters for many words requires further confirmation. For example, "using Pyramidal implementation" in line 193, "using Reference features" in line 195, " the Arbitrary points" in line 389, "The Point measurement" in line 465, etc.
- There should be a standard for using bold letters throughout the text—for example, 𝐀𝐱+𝐁𝐲+𝐂=𝟎 in line 392.
- Incorrect use of root in Eq.12.
- Figure 6 is the same as the Zhihu website (https://zhuanlan.zhihu.com/p/63179478). I do not know if the author has its copyright. In fact, this figure has some shortcomings, such as the representation of points. Therefore, I suggest the author redraw this image.
- Eq. 23 is not written properly and has poor readability.
- There are garbled characters in Line 56.
Comments on the Quality of English Language
Some sentences are too long; There is excessive misuse of capital letters.
Author Response
The authors thank the reviewer for the valuable feedback on the submitted manuscript. The responses are submitted below:
Comment 1: In the flowchart in Figure 1, some modules (EKF Estimator, Initialization, and Zero Velocity Detection) are not connected. Readers cannot determine which stage these modules are used in the flowchart. Some readers may also think this is a flowchart module description.
Response 1: Agree. Thank you for pointing it out. The proposed method is developed on the existing framework of OpenVINS, which provides the estimates of the sliding window of camera poses and reference features to the proposed method for depth map estimation. The proposed method does not change the existing framework of OpenVINS. Therefore, the flowchart in Figure 1 has been updated in the revised manuscript to clarify the proposed method for readers.
Comment 2: Figure 3 needs to be redrawn because the text in the flowchart has been deformed and some text is partially obscured.
Response 2: Agree. Figure 3 has been redrawn.
Comment 3: The ellipsis used in Eq.(4) is inappropriate, and the formula must be written entirely. Other formulas have the same problem, and we hope the author can supplement the omitted formulas.
Response 3: Agree. The revised manuscript has updated the omitted formulas in Eq.(4). The formula in Eq. (19) represents the stacked measurements. It has been mentioned in such a way to describe the concept of measurement stacking.
Comment 4: The use of secondary foot tags is not highly recommended. It would be even better if the variable symbols in the text could be redesigned.
Response 4: Agreed. The suggestion is appreciated. The authors will try their best to accommodate this suggestion. However, an example to address this comment would be appreciated.
Comment 5: Comparing Figure 4 and Figure 5, it is suggested that the first letter of the explanatory phrases in these figures should be capitalized consistently.
Response 5: Agreed. The suggestion is incorporated in the revised manuscript.
Comments 6: In the entire text, the use of capital letters for many words requires further confirmation. For example, "using Pyramidal implementation" in line 193, "using Reference features" in line 195, " the Arbitrary points" in line 389, "The Point measurement" in line 465, etc.
Response 6: Agreed. The suggestion is incorporated in the revised manuscript.
Comments 7: There should be a standard for using bold letters throughout the text—for example, ??+??+?=? in line 392.
Response 7: Agreed. The suggestion is incorporated in the revised manuscript.
Comments 8: Incorrect use of root in Eq.12.
Response 8: Agree. The equation has been corrected.
Comments 9: Figure 6 is the same as the Zhihu website (https://zhuanlan.zhihu.com/p/63179478). I do not know if the author has its copyright. In fact, this figure has some shortcomings, such as the representation of points. Therefore, I suggest the author redraw this image.
Response 9: Agree. The figure has been updated.
Comments 10: Eq. 23 is not written properly and has poor readability.
Response 10: Agree. The explanation has been improved in the revised manuscript.
Comments 11: There are garbled characters in Line 56.
Response 11: Agreed. The issue has been addressed in the revised manuscript.
Reviewer 3 Report
Comments and Suggestions for Authors
In this manuscript, a new technique for generating a depth map from images by tracking Pixel-Points using reference features is proposed. In particular, the proposed method is based on a monocular visual-inertial odometry based algorithm for estimating the image depth map.
The main question addressed by the research presented in this manuscript is the improvement of traditional tracking for real-time applications (e.g. obstacle detection, path planning), which relies on features that may be limited by computational power. The topic is original and relevant to the field, and in the introductory section the authors have clarified the research gap that this manuscript aims to address: Inadequate depth estimation in texture-less environments; ineffectiveness of existing tracking methods based on optical flow and descriptors; computational limitations in real-time requirements; the need to develop a comprehensive tracking algorithm to overcome the challenges of existing traditional techniques.
Regarding the methodology: A comprehensive description of the methodology is provided by the authors as well as graphical representation of the proposed approach. The empirical validation and evaluation of the proposed approach is carried out through extensive experiments on three sequences of publicly available EuRoC dataset. The results obtained are presented, analyzed and discussed in detail. Furthermore, the proposed method is compared with other relevant methods such as Pyramidal LK Optical flow method, based on relevant parameters (estimated depth).
Conclusions are consistent and supported by the experimental results as well as a comparison with other relevant methods.
The authors have presented their work well from both a practical and theoretical point of view. There are 9 figures and 4 tables. The work is technically sound and the references given by the authors are applicable and relevant, there are 25 citations.
Please consider the following corrections and comments:
Please check the keywords, I assume it should be "depth map", but there is only "depth".
Please explain the choice of datasets in more detail, including the reasons why they are suitable for this task. Why is the outdoor dataset KITTI not used for the evaluation of the proposed method?
Regarding tables and figures: Please correct Figure 9, the (d) is missing. There are several panels, so they should be listed as follows: (a) Description of what is contained in the first panel; (b) Description of what is contained in the second panel ... (f) Description of what is contained in the last panel. The same applies to Figure 8. Please check all figure captions to ensure that they match the journal template.
Regarding references: The references given by the authors are applicable and relevant, there are 25 citations, although there are some outdated references, e.g. 1981 and 1997. Please correct the following “[Error! Reference source not found.,2]” all citations should correspond to the journal's template. The referencing style should be consistent throughout the manuscript and based on the journal's template.
Please check proofreading and English spelling, there are some typos and punctuation errors.
Comments on the Quality of English LanguagePlease check proofreading and English spelling, there are some typos and punctuation errors.
Author Response
The authors are thankful to the reviewer for the valuable feedback on the submitted manuscript. The responses are submitted below:
Comment 1: Please check the keywords, I assume it should be "depth map", but there is only "depth".
Response: Agree. The suggestion is incorporated in the revised manuscript.
Comment 2: Please explain the choice of datasets in more detail, including the reasons why they are suitable for this task. Why is the outdoor dataset KITTI not used for the evaluation of the proposed method?
Response 2: Section 6 presents the experimental results using open-source visual-inertial datasets supported by OpenVINS. Finally, Section 7 concludes and discusses future work. We agree with this comment and have addressed it in several places in the revised manuscript, such as lines 49, 171, and 673. The proposed method is built on the existing OpenVINS framework, which mainly supports the visual-inertial datasets. This is why we chose to test our method on the indoor dataset.
Comment 3: Regarding tables and figures: Please correct Figure 9, the (d) is missing. There are several panels, so they should be listed as follows: (a) Description of what is contained in the first panel; (b) Description of what is contained in the second panel ... (f) Description of what is contained in the last panel. The same applies to Figure 8. Please check all figure captions to ensure that they match the journal template.
Response 3: The correction is incorporated in the revised manuscript.
Comment 4: Regarding references: The references given by the authors are applicable and relevant, there are 25 citations, although there are some outdated references, e.g. 1981 and 1997. Please correct the following “[Error! Reference source not found.,2]” all citations should correspond to the journal's template. The referencing style should be consistent throughout the manuscript and based on the journal's template.
Response 4: The correction is incorporated in the revised manuscript.
Comment 5: Please check proofreading and English spelling, there are some typos and punctuation errors.
Response 5: Agree. The proofreading is performed to address spelling and punctuation mistakes.
Round 2
Reviewer 1 Report
Comments and Suggestions for Authors
I appreciate the authors’ responses.
The paper still presents a large quantity of grammatical errors and the english style is not suitable.
In addition, there are some incoherent sentences (line 56, “The study of monocular visual navigation gained interest long ago for several reasons, such as its small size, low cost, and rich environmental information.”), and statements that are not true (line 132, “For this purpose, the traditional SLAM and present-day deep learning methods are not feasible.”). In fact, the current vision-based Advanced Driver Assistance Systems (ADAS) use machine learning methods, including deep learning ones.
All this makes the article unpublishable.
I don’t want to discourage the authors. In my opinion, they have to carefully redesign and rewrite the paper. Not quickly in a couple of days.
Make sure that the introduction (state of the art) focuses on the existing MSCKF for detecting and tracking objects, and the proposed method.

Author Response
Dear,
Thank you very much for your critical observations. The authors acknowledge the distinguished review team's intentions to improve the manuscript so that the ideas presented in the paper can be conveyed well to the audience. The authors have revised the introduction, putting more focus on the state-of-the-art methods presented recently and the proposed method.
Thanks, and regards
Reviewer 2 Report
Comments and Suggestions for Authors
Through careful examination, I found that the main issues I was concerned about have been well addressed, and the quality of the paper has been significantly improved. I have no further modification suggestions.
Author Response
The author thanks the reviewer for his time and effort and appreciates his advice for improving the paper.